# Structure-preserving Machine Learning of Dynamical Systems: A Case for Smaller Models

## Abstract

Dynamical systems naturally evolve on structure-rich manifolds, yet naive machine learning models learn dynamics in flat Euclidean embeddings. This mismatch forces models to *implicitly* learn geometric constraints, resulting in data-intensive training and limited generalization across operating conditions. In this work, we demonstrate how leveraging geometry-informed inductive biases reduces the dependency on larger models to achieve robust generalisation. We investigate a dissipative and a conservative system as use-cases. In the dissipative case, we *identify* a 2-dimensional heat transfer system using a linear state-space formulation where the state operator is constrained to be symmetric positive definite via Riemannian optimization. In the conservative case, we *model* an 18-dimensional Fermi-Pasta-Ulam-Tsingou (FPUT) system on its native symplectic manifold using a symplectic Hamiltonian neural network (SHNN). In the latter case we reveal how structurally-naive models suffer from energy drift when referenced against the true energy surface leading to fragile roll-out generalization, unlike SHNNs which conserve phase-space volume along the correct energy level.

## 1 Introduction

Most real-world physical systems are governed by underlying dynamical systems. Whether modeling from first principles or learning from data, we can reasonably assume the system temporally evolves on some lower-dimensional manifold embedded in a high-dimensional physical coordinate representation. This manifold can be described as a generalized space whose geometric structure is time-invariant thereby preserving the system's structural properties (symmetries, invariances, conservation laws) even as trajectories are observed at different temporal or parametric scales. Thus, imposing structure-preserving inductive biases in machine-learning models that operate on such spaces improves generalization and reduces reliance on large models and volumes of data to implicitly recover the underlying structure. This is a necessary path forward for modeling real-world physical systems across engineering domains.

In this paper, we reinforce this claim with a comparative study of structure-preserving versus structurally naive approaches on one dissipative system and one conservative system. In the former use-case, we present a system identification (SID) of a dynamical system via a structure-preserving, linear matrix model to learn the phase-space dynamics of a 2-dimensional heat transfer system, while in the latter, an established structure-preserving neural-network architecture is adopted to learn the conservative dynamics of an 18-dimensional system. Our overarching aim is to illustrate how superior generalization and stability can be achieved with smaller, yet structure-aware, models.

### 1.1 Physics-informed Biasing

Encoding prior knowledge into machine learning through physics informed-biases is a growing research topic aimed at improving training parsimony and generalization. Most popular are physics-informed neural networks (PINNs) which incorporate constraints and laws directly into the learning via the loss function (Raissi et al., 2020). PINNs overcome the computational expense often encountered with numerical solvers when solving forward and inverse problems for partial differential

equations (PDEs) for high-dimensional and non-linear applied problems, for example Raissi et al. (2019), Jagtap et al. (2022), Berardi et al. (2025).

Traditionally, PINNs encode physics through loss penalties on the PDE residual and boundary conditions (BCs), rather than by encoding inductive biases into the model architecture. Consequently, the residual-based learning still conditions generalization on the representativeness of the training data, sampling, and the implicit regularization of stochastic optimization, where the optimizer resides in flat Euclidean parameter space and is not aware of the non-Euclidean structure of the phase-space (e.g., symplectic form) (Zhang et al., 2017). Once trained, structurally-naive models often struggle to extrapolate to unseen initial conditions or parameters. This has motivated structure-preserving approaches such as Hamiltonian/symplectic neural networks (HNNs/SHNNs) (Greydanus et al., 2019; David & Méhats, 2023) and SympNets (Jin et al., 2020) for learning conservative systems, symmetry-equivariant neural networks (Wang et al., 2022) and thermodynamics-informed for learning non-conservative systems (Hernández et al., 2023; Barbaresco, 2022), and other structure-oriented inductive-biasing approaches for the data-driven discovery of intrinsic dynamics (Champion et al., 2019; Floryan & Graham, 2022).

We argue that extensive knowledge of structure-rich spaces from classical mechanics and differential geometry remains an underutilized opportunity for developing data-driven models that can generalize at a structural level rather than merely fitting data with predefined constraints.

## 2 GEOMETRIC UNDERPINNING OF PHYSICAL SYSTEMS

Dynamical systems underlying physical phenomena can be broadly classified by their energy behavior: some conserve energy (conservative), while others dissipate it (dissipative). All conservative systems can be represented in a Hamiltonian formulation whose dynamics evolve on a symplectic manifold, preserving volume form as they flow through phase space. On the other hand, dissipative systems lack symplectic structure; their natural geometries depend on the choice of model formulation. For example, when dynamics converge to stable, low-dimensional (non-chaotic) attractors, models such as linear state-space matrix representations are suitable, where the gradient flows on Riemannian manifolds.

Conceptually, one can view mathematical models of dynamics as living in a hierarchy of increasingly general geometric spaces, where parameter perturbations deform the vector field and are observed as changes in flow trajectories on an underlying manifold (Figure 1). Most governing laws for physical systems are smooth, allowing us to work on smooth manifolds with well-defined tangent spaces and differentiable maps (see, Asselmeyer-Maluga & Brans 2007). We illustrate these ideas with the following two use-cases.

### 2.1 DISSIPATIVE USE-CASE: HEAT TRANSFER SYSTEM

The conduction dynamics of a material system undergoing one-dimensional heat transfer laterally along its thickness can be described by:

$$\frac{\partial u(x,t)}{\partial t} = \frac{k}{\rho c}\frac{\partial^2 u(x,t)}{\partial x^2} + \frac{q(x,t)}{\rho c}, \tag{1}$$

where: $k$ is the thermal conductivity of the material ($W/mK$), $u(x,t)$ is the temperature as a function of spatial coordinate $x$ and time $t$, $\rho$ is the density of the material ($kg/m^3$), $c$ is the specific heat capacity of the material ($J/kgK$), $k/\rho c$ is the thermal diffusivity and $q(x,t)$ is the internal heat generation per unit volume ($W/m^3$).

Considering the discrete nature of physical systems, we must approximate the continuous temperature field in 1 by reducing it to a discrete domain. We adopt a linear time-invariant state space model (LSSM) approach whose matrix representation offers a compact formulation while preserving the geometric structure that governs their evolution. We assume a discrete approximation of the material system as $m$=2 temperature states $T_{ext1}, T_{ext2} \in$ T (on either side of the material thickness) and an external forcing $T_{ext}$ which represents the ambient temperature influencing the dynamics through convection, directly influencing $T_{ext1}$ only. These are represented in a continuous-time LSSM formulation as follows:

$$\frac{d\mathbf{T}}{dt} = A\mathbf{T} + B\mathbf{U} = \begin{bmatrix} \dfrac{-U_{ext1,ext2}}{C_{ext1}} & \dfrac{U_{ext1,ext2}}{C_{ext1}} \\ \dfrac{U_{ext1,ext2}}{C_{ext2}} & \dfrac{-U_{ext1,ext2} - U_{ext2,ext}}{C_{ext2}} \end{bmatrix} \begin{bmatrix} T_{ext1} \\ T_{ext2} \end{bmatrix} + \begin{bmatrix} 0 \\ \dfrac{U_{ext2,ext}}{C_{ext2}} \end{bmatrix} [T_{ext}] \quad (2)$$

where, $\mathbf{T} \in \mathbb{R}^2$ is a state vector, $\mathbf{U} \in \mathbb{R}^{2 \times 1}$ is input vector, $A \in \mathbb{R}^{2 \times 2}$ contains the information about the unforced dynamics of all system states $\mathbf{T}$ while $B \in \mathbb{R}^{2 \times 1}$ determines how the input matrix $\mathbf{U}$ (forcing) influences the states $\mathbf{T}$. Further, $U_{exti,extj}$ is the thermal transmittance $(W/m^2 K)$. The structure of $A$ reflects the physical topology of the discretized domain via the lumped parameter approach as described in Xuereb Conti et al. (2023) where, where $A = V \Lambda \mathbf{T}^{-1}$, $\Lambda = \text{diag}(\lambda_1, \lambda_2, \dots, \lambda_N)$ is the diagonal matrix of eigenvalues and $V$ is the matrix of eigenvectors of $A$. Let $\mathbf{T} \in \mathbb{R}^2$ be the state vector and $\mathbf{U} \in \mathbb{R}^{1 \times 1}$ the input vector. $A$ remains invariant to the order of the system provided the topological connectivity between the states is preserved.

In order to solve the system in 2, we must convert time-continuous $A$ to discrete-time $\Phi_A$ via the matrix exponential expansion, as follows:

$$\Phi_A = e^{A\tau} \text{ and } \Phi_B = A^{-1}(e^{A\tau} - I)B \tag{3}$$

where $\tau$ is the time-step for discretization. Further expansion of 16 can be found in Appendix A.

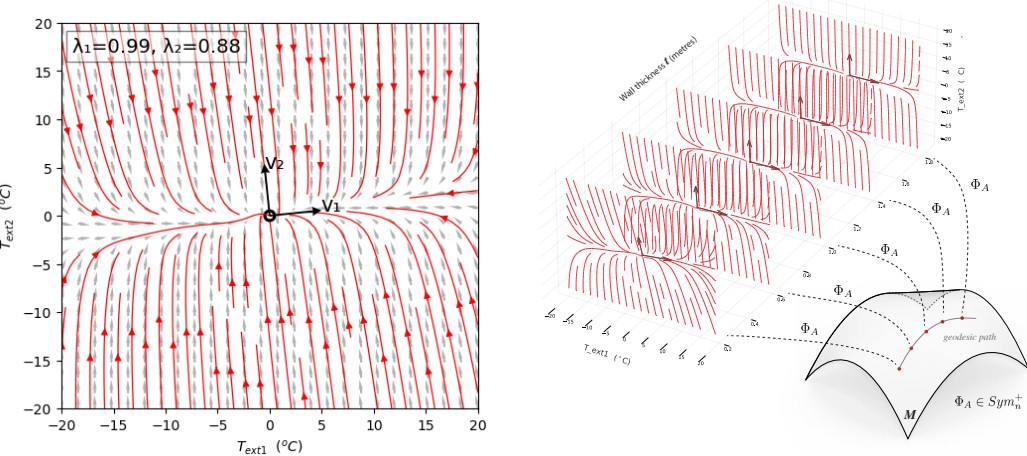

(a) Phase space geometry of the heat transfer dynamics for a 200 millimeter material thickness.

(b) Phase portraits on the symmetric positive definite manifold.

Figure 1: Comparison of (a) heat transfer phase space and (b) SPD manifold portraits.

### 2.1.1 STABLE GENERALIZATION ON THE SYMMETRIC POSITIVE DEFINITE MANIFOLD

Formulating dynamical systems in the state space matrix format offers advantages and opportunities for preserving natural structure related to geometry governed by the invariant and symmetry features of the matrices. In several instances, the formulation of system matrix $A$ in equation 2 belongs to the symmetry matrix manifold $Sym_n$ where $A = A^T$ and which is a Euclidean (flat) subspace of the space of all matrices $\mathbb{R}^{n \times n}$. Its time-discretization $\Phi_A$ belongs to the symmetric positive definite (SPD) manifold $Sym_n^+$ which is a non-Euclidean space (curved) and a submanifold of $Sym_n$ where matrices are symmetric but specifically, positive definite. For a matrix to be positive definite, all its eigenvalues must be positive (i.e. $Re(\lambda_i) > 0) \forall i$). The SPD manifold $\mathcal{M}$ is a smooth differentiable topological space equipped with an invariant Riemannian structure (i.e. Riemannian manifold). The structure facilitates a Riemannian metric that varies smoothly from point to point where every point is equivalent to a unique and valid physical system. For further reading on Riemannian metrics, see Sommer et al. (2020). For each system matrix $\Phi_A \in \mathcal{M}$, it is possible to compute a tangent

space $\mathcal{T}_{\Phi_A}\mathcal{M}$. The operator used to map from a point on the manifold to its tangent space is given by the logarithmic map $\mathrm{Log}_{\Phi_A}(m) : \mathcal{M} \to \mathcal{T}_{\Phi_A}\mathcal{M}$ while the inverse is given by the exponential map $\mathrm{Exp}_{\Phi_A}(m) : \mathcal{T}_{\Phi_A}\mathcal{M} \to \mathcal{M}$. Therefore, we can interpret the tangent space at a given system $\Phi_A$ in equation 2 on the $Sym_n^+$ manifold $\mathcal{M}$ as the linearized space of all possible infinitesimal perturbations of $\Phi_A$ that preserve the symmetric structure (1).

Thus, when interpreted geometrically, the latter is equivalent to 3 implying that time-discretization of $A$ is a projection from continuous-time dynamics residing in the Euclidean space of all possible symmetric dynamical systems $Sym_n$ to the non-Euclidean space $Sym_n^+$ of symmetric but stable discrete-time dynamical systems, by means of their positive eigenvalues implying positive definiteness. In further detail, $e^{A\tau}$ is a bilinear map that geometrically maps the complex $s$-plane to the complex unit circle in the $z$-plane where system stability is preserved by wrapping the stable eigenvalues located in the left half-plane (i.e., $Re(\lambda_i) < 0$)) within the unit circle in the $s$-plane where $Re(\lambda_i) > 0$). System matrices $\Phi_A$ that lie on the surface of the SPD manifold are positive *semi-definite* attributed to their low-rank and are said to be *bistable* due to some eigenvalues $Re(\lambda_i) \geq 0$). In general, as you move towards the boundary in the stratified space composing the SPD manifold, the matrix loses rank, meaning that fewer independent eigendirections remain for the system trajectories to evolve in. For further reading on the role of symmetry in dynamical systems, see Marsden & Ratiu (2013).

### 2.1.2 SPD-AWARE MACHINE LEARNING VIA RIEMANNIAN OPTIMISATION

Having illustrated the geometric connection between the phase space underlying physical descriptions of dynamical systems, it becomes natural to leverage these geometric representations for preserving structure when machine learning phase space dynamics from data. Assuming the availability of measurement data from the system, our mission is to uncover the underlying eigenstructure that governs a measured system's behavior, by perturbing or 'nudging' the physics-approximated state space model within the underlying $SPD$ manifold, closer towards a dynamical system that represents the stable dynamics underlying the measured temperature data. In the control/dynamical systems community, this could be interpreted as manifold-constrained system identification (SID).

We start with an initial state matrix $A$ that is derived from Physics but misspecified (see Table 3), and which is used as an initial guess at the start of the optimization. Physical systems are measured at discrete time-steps, hence it is necessary to reformulate equation 2 into its discrete-time form via equation 3, as follows:

$$\mathbb{T}_{t+1} = \Phi_A \mathbf{T}_t + \Phi_B \mathbf{U}_t. \tag{4}$$

The optimization goal is to learn a new LSSM whose matrices, denoted $\hat{\Phi}_A$ and $\hat{\Phi}_B$, better fit the target measurement data. The matrices are parameterized as tensors of size $n \times n$ and $n \times m$, respectively. The optimization problem described above may be stated as:

$$\hat{\Phi}_A, \hat{\Phi}_B = \underset{\Phi_A, \Phi_B}{\arg\min} \, \mathcal{J}(X|\Phi_A, \Phi_B), \tag{5}$$

$$\text{s.t. } \Phi_A^\top = \Phi_A \text{ and } \mathbf{T}^\top \Phi_A \mathbf{T} > 0 \, \{\mathbf{T}|\mathbf{T} \in \mathbb{R}^2\}, \tag{6}$$

where the loss function, $\mathcal{J}$, is defined as:

$$\mathcal{J}(X|\Phi_A, \Phi_B) = \sum_{i=1}^{n-1} \left\| \Phi_A \mathbf{T}_i + \Phi_B \mathbf{T}_i - \mathbf{T}_{i+1} \right\|_2^2. \tag{7}$$

To preserve stability of $\Phi_A$ via the symmetric positive structure, we adopt the Riemannian adaptive optimization method (RAdam) Bécigneul & Ganea (2019) to estimate the $\hat{\Phi}_A$ tensor where gradient updates follow the curved geodesic. The Riemannian gradient is given by:

$$\nabla_{\Phi_A} \mathcal{J}(\Phi_A^{(i)}), \tag{8}$$

and therefore, gradient updates follow the curved geodesic by projecting the gradient onto the tangent space as follows:

$$\Phi_A^{(i+1)} = \exp_{\Phi_A^{(i)}}\left(-\eta.\frac{\hat{m}_i}{\sqrt{\hat{v}_i}}\right), \tag{9}$$

where where $\eta$ is a user determined learning rate, $\hat{m}_i$ and $\hat{v}_i$ are the bias-corrected first and second moment estimates, respectively which summarise the history of gradients of $\mathcal{J}$ to inform the adaptive direction of the geodesic update. The RAdam was implemented using the `geoopt` Python library (Kochurov et al., 2020). On the other hand, gradient updates for learning $\hat{\Phi}_B$ were computed in flat Euclidean space using Adam (Kingma & Ba, 2017) in `torch.optim` Python library (Paszke et al., 2019). In an alternative approach, $\hat{\Phi}_A$, may also be parameterized by the lower Cholesky decomposition via $\hat{\Phi}_A = LL^T$ to ensure optimization stays within the SPD manifold.

## 2.2 CONSERVATIVE USE-CASE: FERMI-PASTA-ULAM-TSINGOU SYSTEM

The FPUT chain (Fermi et al., 1955) provides a classic benchmark for studying nonlinear dynamics in many-body systems. It models a set of particles connected by springs, where nonlinearity arises from higher-order terms in the spring potential. We consider a fixed-end chain of $N$ masses, with $M = N - 1$ interior degrees of freedom, leading to a $2M$-dimensional canonical phase space $z = (q_1, \ldots, q_M, p_1, \ldots, p_M)$.

The Hamiltonian of the cubic FPUT–$\alpha$ model is:

$$H(q,p) = \sum_{i=1}^{M} \frac{1}{2}p_i^2 + \sum_{i=0}^{M} \frac{1}{2}(q_{i+1} - q_i)^2 + \frac{\alpha}{3}(q_{i+1} - q_i)^3, \qquad q_0 = q_N = 0, \tag{10}$$

where $q_i$ and $p_i$ denote displacement and momentum of the $i$th mass, respectively, and $\alpha$ controls the nonlinear stiffness. For $\alpha = 0$, the system reduces to a linear chain with nearly elliptical phase portraits, while $\alpha \neq 0$ produces asymmetric level sets (e.g. the 'teardrop' shapes in Figure 2). Here, $q_0 = q_{M+1} = 0$.

The corresponding equations of motion follow from Hamilton's equations,

$$\dot{q}_i = \frac{\partial H}{\partial p_i} = p_i, \qquad \dot{p}_i = -\frac{\partial H}{\partial q_i} = q_{i+1} - 2q_i + q_{i-1} + \alpha[(q_{i+1} - q_i)^2 - (q_i - q_{i-1})^2]. \tag{11}$$

where, $i = 1, \ldots, M$. In compact form, the Hamiltonian flow can be expressed as:

$$X_H(z) = J\nabla H(z), \qquad J = \begin{bmatrix} 0 & I \\ -I & 0 \end{bmatrix}, \tag{12}$$

where the canonical matrix $J$ defines the symplectic structure. The associated two-form

$$\omega = \sum_{i=1}^{M} dq_i \wedge dp_i, \tag{13}$$

is exactly preserved under the dynamics. This invariance guarantees conservation of phase-space volume (Liouville's theorem) and energy, $H(q,p)$.

The Hamiltonian $H$ can be viewed as a time-invariant surface defined over the phase space. While the full 18-dimensional energy surface over the full phase space cannot be visualized, in Figure 2 (bottom row) we illustrate two-dimensional projection slices through $H$, evolving over time alongside the trajectory of the first coordinate pair $(q_4, p_4)$ evolving over time ($t = 0, \ldots, 150$). A slice at each time $t$, is achieved by varying $(q_4, p_4)$ while holding all other coordinates fixed at their instantaneous values. In the plots, the white contour is the level set at which the invariant energy hypersurface intersects with the $(q_1, p_1)$–slice at time $t$. The energy level is time-invariant and only appears to change due to the sliced view of the 18-dimensional surface.

Since Hamiltonian flows satisfy $X_H(z) = J\nabla H$, the flow is *tangent* to the (constant) energy level set in ever slice, as can be seen in Figure 2. Importantly, when a trajectory is visibly *jumping* between level sets in these plots, it is indicative of *energy drift* (non–conservation) arising from model discrepancy from the true energy surface.

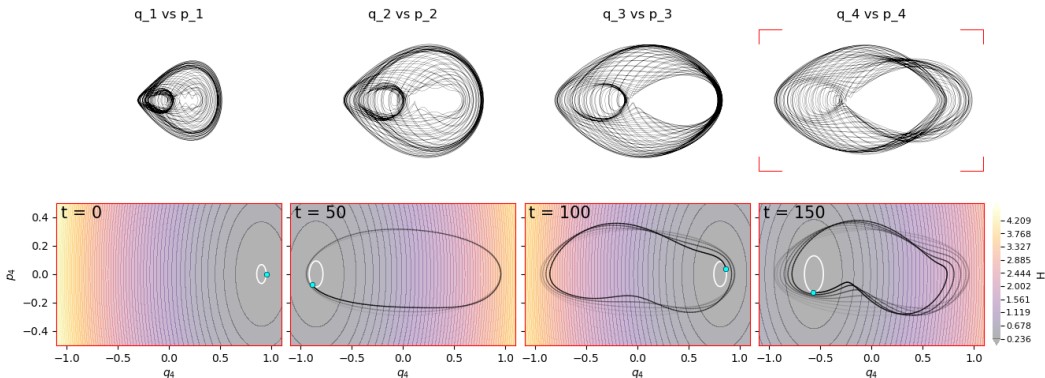

Figure 2: Top: 2D phase space projections of the first 4 out of 9 conjugate state pairs, from the full 18-dimensional phase space geometry of the FPUT system where $\alpha = 0.25$. Bot: 2D snapshots of the $p_4, q_4$ flow evolving on the 2D-projected Hamiltonian set and corresponding energy level (white).

### 2.2.1 SYMPLECTIC-AWARE MACHINE LEARNING VIA SHNNS

The symplectic structure governing conservative systems can be leveraged to learn the conservative dynamics of Hamiltonian systems from data. Hamiltonian neural networks (HNNs) (Greydanus et al., 2019) take physical coordinates $(q, p)$ as input and learn a single scalar Hamiltonian $H_\theta(q, p)$; the dynamics are obtained via the symplectic gradient $f_\theta(z) = J\nabla H_\theta(z)$ with $z = (q, p)$. Specifically, we use symplectic Hamiltonian neural networks (SHNN), which extend HNNs using a symplectic time discretization via the implicit midpoint rule, while retaining the Hamiltonian parameterization (David & Méhats, 2023). This setup ensures the learned vector field is Hamiltonian by construction and that the roll-out map is symplectic due to the integrator, promoting energy and structure preservation.

## 3 EXPERIMENTS AND RESULTS

### 3.1 DISSIPATIVE USE-CASE

A sequence of one year's worth of synthetic, hourly measurement temperature data T (8,759 hours, $\tau = 1$) where $T \in \mathbb{R}^{8759 \times 1}$ was generated via a high-fidelity numerical analysis of a homogeneous material system using EnergyPlus [1]. The selected physical and thermodynamic properties are found in Table 3, while the ambient dry bulb temperature acting as a forcing $U \in \mathbb{R}^{8759 \times 2}$ was obtained from a historical weather file located in London and Chicago (ladybug tools, 2013), and used as an input for the numerical simulation. While the former was split for testing/training, the latter was used as secondary test set for testing out of distribution initial conditions. The data was obtained from an earlier study in Xuereb Conti et al. (2023).

To highlight the benefit of leveraging structure-informed biasing, we repeated the same modeling task across three popular time-series modeling approaches built without structure-awareness, namely: Random forest (RF), extreme gradient boosting (XGBoost), and long short-term memory networks (LSTMs). Additionally, we repeat the system identification of the linear state space model where $\hat{\Phi}_A$ and $\hat{\Phi}_B$ tensor elements are learned using only Euclidean gradient updates, denoted as EucOpt, rather than through the proposed Riemannian optimization scheme, which we denote RieOpt.

---

[1]EnergyPlus is a simulation software adopted widely in the thermal energy modeling community NREL (2017)

| Method | $T_{ext1}$ | | $T_{ext2}$ | |
| --- | --- | --- | --- | --- |
| | London | Chicago | London | Chicago |
| LSSM from Physics ($\Phi_A, \Phi_B$) | 2.86e+00 | 1.07e+01 | 6.06e-01 | 2.10e+00 |
| RieOpt ($\hat{\Phi}_A, \hat{\Phi}_B$) | **4.00e-01** | **1.36e+00** | 5.07e-01 | **1.79e+00** |
| EucOpt ($\hat{\Phi}_A, \hat{\Phi}_B$) | 1.28e+00 | 3.35e+00 | 5.80e-01 | 1.98e+00 |
| RF | 6.81e-01 | 2.41e+01 | 2.32e-01 | 1.63e+01 |
| XGBoost | 5.02e-01 | 2.23e+01 | **1.06e-01** | 1.33e+01 |
| LSTM | 2.57e+01 | 4.01e+01 | 6.10e+00 | 7.85e+00 |

Table 1: $MSE$ error of models applied to test datasets.

### 3.1.1 RESULTS

While Figure 5 suggests that the structure-naive models (RF, XGBoost and LSTM) seem to roll-out the test segments accurately, as evidenced by their mean square error (MSE) loss for the unseen time-steps in Table 1, their training convergence is significantly slower as can be noted on comparing Figure 8 with the structure-preserving EucOpt and RieOpt in Figure 7.

To evaluate generalisation for unseen initial conditions, we expose all trained models to an unseen sequence of hourly forcing temperatures $T_{ext}$ across one year, located in Chicago (ladybug tools, 2013) where temperatures exhibit different seasonal extremes to London (Figure 6). The underlying thermal dynamics of the material system is invariant of the forcing, implying that if a model has successfully captured the relationship between the forcing and the unforced dynamics, it should generalize for the unseen initial condition. Observing both the $MSE$ loss in Table 1 and the time-series fit when predicting the indirectly forced state $T_{ext1}$ in Figure 5 we can instantly note how the structurally-naive approaches demonstrate instability whereas, RieOpt and EucOpt demonstrate global stability in capturing the dynamics, in particular the former, as illustrated by the nudged phase portrait in Figure 5 (bottom, left). The structure-aware approach has learned the phase space vector field decoupled dynamics, as opposed to the investigated model free approaches, that learn the forced response of the system as a time series.

### 3.2 CONSERVATIVE USE-CASE

Training data were generated by integrating the Hamiltonian FPU-$\alpha$ system 10 with the symplectic leapfrog (Störmer–Verlet) scheme, which preserves the symplectic form and offers good long-time energy behavior. We simulate a single long trajectory $Z$ ($t = 30,000$ steps, $\tau = 0.1$) with fixed-end boundary conditions, initialized by exciting the first normal mode, $q_i(0) = \sin\left(\frac{i\pi}{N}\right)$ and $p_i(0) = 0$ for $i = 1, \ldots, N - 1$. The resulting time series ($Z \in \mathbb{R}^{30000 \times 18}$) is split chronologically into 80/20% for training/testing ($Z_{tr} \in \mathbb{R}^{24000 \times 18}$ / $Z_{te} \in \mathbb{R}^{6000 \times 18}$). We benchmark structure-preserving SHNNs against a naive LSTM and a NeuralODE (Chen et al., 2019) baseline. LSTM and NeuralODE inputs $(p, q)$ were standardized using the training split mean $\mu$ and standard deviation $\sigma$ and de-standardized for evaluation. The same $\mu, \sigma$ were applied to the test split and to any evaluations on unseen initial conditions. SHNN models were trained directly in physical coordinates to preserve the canonical symplectic structure. To ensure fairness, all metrics used to compare models are computed in physical units.

For SHNNs and NeuralODEs we sweep over the number of hidden layers $L \in \{n_f, 2n_f, 4n_f, 8n_f\}$ and hidden widths $W \in \{n_f, 2n_f, 4n_f, 8n_f\}$, where $n_f$ denotes the dimension of the dynamical state. For the LSTM, we sweep over $W$ only. Each model is trained for $2,000$ epochs with the Adam optimizer (learning rate $3 \times 10^{-3}$). For all models we evaluate: a) the average one-step update (one-step$_{MSE}$) across the test set, b) average autoregressive roll-out ($1,000$ steps) prediction of the test set (roll-out$_{MSE}$), and c) the average drift from the true Hamiltonian (drift$_{RMS}$). The latter is measured by computing $\Delta H_k = H(\hat{z}_{t+k}) - H(\hat{z}_t)$ where each model is autoregressively rolled out for $1,000$ steps from the initial state $(q_0, p_0)$ and where $\hat{z}_{t+k}$ is the predicted state. The drift $MSE$ is obtained as the mean loss from the true Hamiltonian (intial state) across the roll-out. The drift $MSE$ provides implicit insight about the stability for longer horizon predictions where, a large drift implies that energy levels on the energy surface are being crossed and thus, the total energy is not conserved.

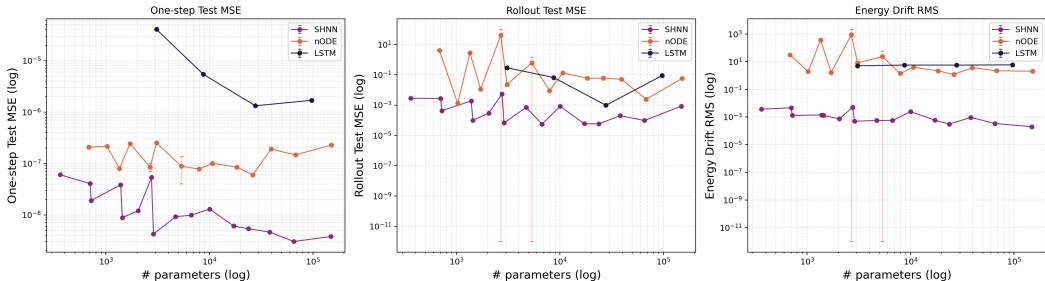

Figure 3: One-step $MSE$ (left), test roll-out $MSE$ (centre) and energy drift $RMS$ (right) for varying model sizes across SHNN, NeuralODE and LSTM.

Table 2: One-step test loss and energy drift loss for varying model sizes across SHNN, NeuralODE, and LSTM. Hand-picked 'best' size vs. loss trade-off models in bold.

| | | **SHNN** | | | **NeuralODE** | | | **LSTM** | | |
|---|---|---|---|---|---|---|---|---|---|---|
| $L$ | $W$ | $\text{Test}_{MSE}$ | $\text{Drift}_{RMS}$ | Params | $\text{Test}_{MSE}$ | $\text{Drift}_{RMS}$ | Params | $\text{Test}_{MSE}$ | $\text{Drift}_{RMS}$ | Params |
| 1 | 18 | 6.045e-08 | 3.697e-03 | 361 | 2.079e-07 | 3.141e+01 | 684 | 4.065e-05 | 5.090e+00 | 3078 |
| 1 | 36 | 1.908e-08 | 1.319e-03 | 721 | 7.991e-08 | 3.775e+02 | 1350 | 5.447e-06 | 5.702e+00 | 8730 |
| 1 | 72 | **8.876e-09** | **1.322e-03** | **1441** | **7.430e-08** | **1.787e+00** | **2682** | 1.329e-06 | 5.687e+00 | 27810 |
| 1 | 144 | 4.256e-09 | 5.035e-04 | 2881 | 5.472e-08 | 1.617e+00 | 5346 | **1.694e-06** | **5.914e+00** | **97074** |
| 2 | 18 | 4.064e-08 | 4.638e-03 | 703 | 2.160e-07 | 1.919e+00 | 1026 | - | - | - |
| 2 | 36 | 1.209e-08 | 7.420e-04 | 2053 | 9.488e-08 | 1.802e+03 | 2682 | - | - | - |
| 2 | 72 | 5.284e-09 | 3.982e-04 | 6697 | 7.794e-08 | 1.420e+00 | 7938 | - | - | - |
| 2 | 144 | 3.901e-09 | 5.654e-04 | 23761 | 5.982e-08 | 1.194e+00 | 26226 | - | - | - |
| 4 | 18 | 2.606e-08 | 9.681e-04 | 1387 | 2.437e-07 | 1.673e+00 | 1710 | - | - | - |
| 4 | 36 | 7.120e-09 | 1.178e-03 | 4717 | 1.229e-07 | 4.533e+01 | 5346 | - | - | - |
| 4 | 72 | 3.574e-09 | 5.463e-04 | 17209 | 1.391e-07 | 1.396e+00 | 18450 | - | - | - |
| 4 | 144 | 3.091e-09 | 3.445e-04 | 65521 | 1.707e-07 | 1.484e+01 | 67986 | - | - | - |
| 8 | 18 | 1.338e-08 | 1.073e-03 | 2755 | 2.221e-07 | 1.206e+00 | 3078 | - | - | - |
| 8 | 36 | 1.302e-08 | 2.453e-03 | 10045 | 1.009e-07 | 3.970e+00 | 10674 | - | - | - |
| 8 | 72 | 4.621e-09 | 9.373e-04 | 38233 | 1.910e-07 | 3.709e+00 | 39474 | - | - | - |
| 8 | 144 | 3.799e-09 | 1.995e-04 | 149041 | 2.296e-07 | 2.028e+00 | 151506 | - | - | - |

### 3.2.1 RESULTS

Figure 3 illustrates the loss for one-step prediction (left panel), free roll-out over $1,000$ steps of the unseen test set (centre panel) and loss for the energy drift (right panel), for varying model sizes across all three models. As expected, increasing the model size improves one-step predictions across all three models but not necessary longer-horizon behaviour as can be seen in the centre panel. Compact symplectic models beat larger, structure-naive baselines on test rollout and drift: most notably, a small SHNN (1,441 params) achieves a significantly better roll-out than the best LSTM (97,074 parameters) which is justified by the lower drift loss, underscoring the benefit of structure Table (2). NeuralODEs vary widely where the best case still drifts significantly more than the SHNN. The impact of enforcing symplectic conservation is especially highlighted in Figure 4a where the overlayed phase trajectory (blue) in the projected phase space $(q_4, p_4)$ remains close to the predicted energy level that aligns well with the true Hamiltonian. We visualise time-evolving snapshots of the trajectory. Further, when rolling out for perturbed unseen initial conditions in Figures 4b and 4c, the smaller SHNN demonstrates better stability than the best performing yet structure-naive LSTM whose trajectory drifts across the energy levels thus, energy is lost. This visualization helps explain why structurally-naive models such as LSTMs tend to generalize poorly for long roll-outs and out-of-distribution conditions.

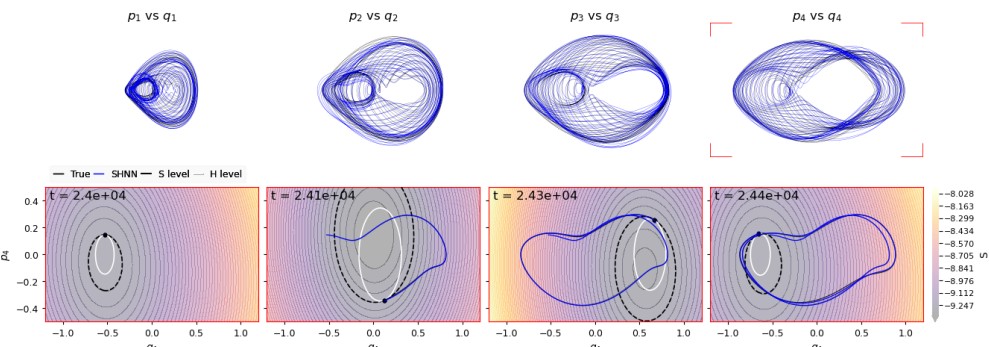

(a) Predicted $(q_4, p_4)$ **unseen test data** trajectory (blue line) via SHNN L=1,W=72(1,441 parameters) trained for 2K epochs. The predicted energy level is represented by the dashed ellipse and can be seen to approximate the true energy level (white ellipse) closely.

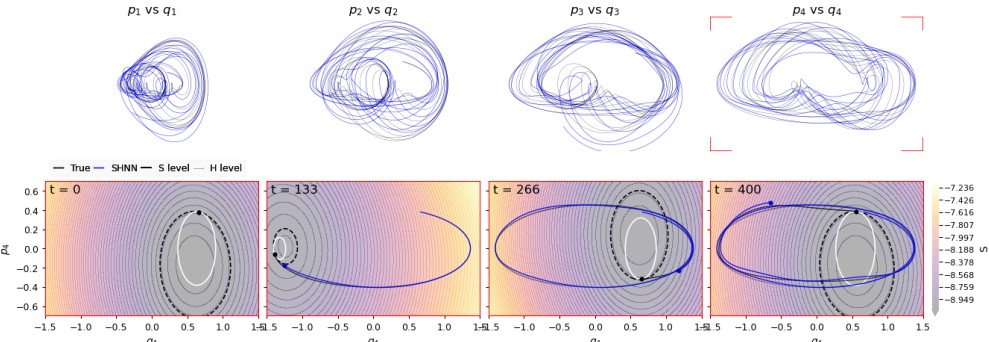

(b) Predicted $(q_4, p_4)$ **unseen initial condition** trajectory (blue line) via SHNN L=1,W=72(1,441 parameters) trained for 2K epochs.

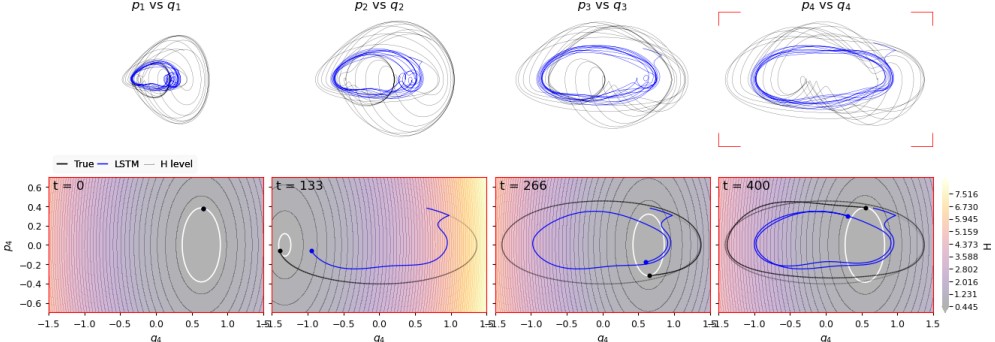

(c) Predicted $(q_4, p_4)$ **unseen initial condition** trajectory (blue line) via LSTM W=144(97,074 parameters) trained for 2K epochs.

Figure 4: Illustrating phase space stability by means of predicted flows on the Hamiltonian.

## 4 CONCLUSION

We showed that structure-aware models can reduce dependence on model size while improving robustness. In two use cases: Riemannian optimization for system identification and symplectic Hamiltonian neural networks for conservative dynamics, varying model size revealed that stable generalization across initial conditions is achievable with models that are much smaller than equally robust, structure-naive baselines. By encoding geometric and physical priors (symmetric positive definite constraints for stable dissipative systems and symplectic structure for conservative systems), we obtain lower long-horizon rollout error and smaller energy drift, even when one-step accuracy alone might suggest simply making models larger.

ETHICS STATEMENT

ChatGPT and Google Gemini were used to polish the writing of the paper. All data and code will be made available in a public repository.

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

## A   APPENDIX: DISSIPATIVE USE-CASE SUPPLEMENTARY DETAILS

State Space Matrix model formulation:

$$\frac{dT_{ext1}}{dt} = \frac{1}{C_{ext1}} \left[ \frac{1}{R_{\text{ext2\_ext1}}} \left( T_{\text{ext1}} - T_{\text{ext2}} \right) \right],$$

$$\frac{dT_{ext2}}{dt} = \frac{1}{C_{ext2}} \left[ \frac{1}{R_{\text{ext2\_ext1}}} \left( T_{\text{ext1}} - T_{\text{ext2}} \right) \right] \tag{14}$$

$$+ \frac{1}{R_{\text{ext\_ext2}}} \left( T_{\text{ext2}} - T_{\text{ext}} \right).$$

Matrix exponential expansion:

$$e^{At} = \mathbf{I} + At + \frac{A^2 t^2}{2!} + \frac{A^3 t^3}{3!} + ... + \frac{A^k t^k}{k!} + ..., \tag{15}$$

which leads to the following equation for the discrete time dynamics:

$$\dot{\mathbf{T}}(t) = e^{At}\mathbf{T}(t_0) + A^{-1}(e^{At} - I)B\mathbf{U}(t_0). \tag{16}$$

Table 3 displays the values of the various physical and thermophysical parameters in the dissipative use-case.

| Genre | Property | Target (measurements) | Misspecification (physics) |
|---|---|---|---|
| Physical properties | volume | 1.8 $m^3$ | 3.6 $m^3$ |
| | layer thickness | 0.2 $m$ | 0.4 $m$ |
| Thermophysical material properties | conductivity | 0.72 $W/mK$ | 0.2 $W/mK$ |
| | density | 1920 $kg/m^3$ | 1920 $kg/m^3$ |
| | specific heat capacity | 780 $J/kgK$ | 780 $J/kgK$ |
| Convection coefficients | outdoor convection | 25 $W/m^2K$ | 20 $W/m^2K$ |
| Theremophysical air properties | air density | 1.2 $kg/m^3$ | 1.2 $kg/m^3$ |
| | air specific heat capacity | 100 $J/kgK$ | 100 $J/kgK$ |

Table 3: Physical and thermodynamic properties used to generate target data via numerical analysis in EnergyPlus and to initialize the state space model optimisation, respectively.

## B  APPENDIX: DISSIPATIVE USE-CASE SUPPLEMENTARY DETAILS

Figure 5 displays the results for model training and testing on the London dataset (top and bottom right). Note that all model-free approaches demonstrate instability in contrast with, the model-based approaches **Rie opt** and **Euc opt** which demonstrate global stability in capturing the dynamics, in particular the former, as illustrated by the nudged phase portrait in the bottom left panel. Figure 6 displays the results of the models when applied autoregressively to the unseen Chicago dataset, which demonstrates different seasonal extremes compared to the London dataset.

Figure 7 displays the MSE loss per epoch of the model-based approaches **Rie opt** and **Euc opt** during training on the London dataset. Note that **Rie opt**, trained by optimizing on the Riemannian manifold, converged significantly faster than the model optimized in Euclidean space.

Figure 8a displays the five-fold cross validated (CV) MSE, for a sweep of forest sizes. The training portion of the London dataset was used to perform this sweep, with a forest size of 250 trees selected for the random forest and 60 for XGBoost. Figure 8b displays the convergence of the investigated LSTM architectures in training. Considerable instability was observed in LSTM training and testing on the London dataset, with best results achieved by learning $T_{ext1}$ and $T_{ext2}$ independently. Both LSTMs used 64 hidden layers, with a window size of 100. There is a strong seasonal variation to both the London and Chicago dataset. The poor performance of the LSTMs was attributed to the relatively small size of the training dataset, which limited the window size and made it difficult for the LSTMs to capture seasonal variations.

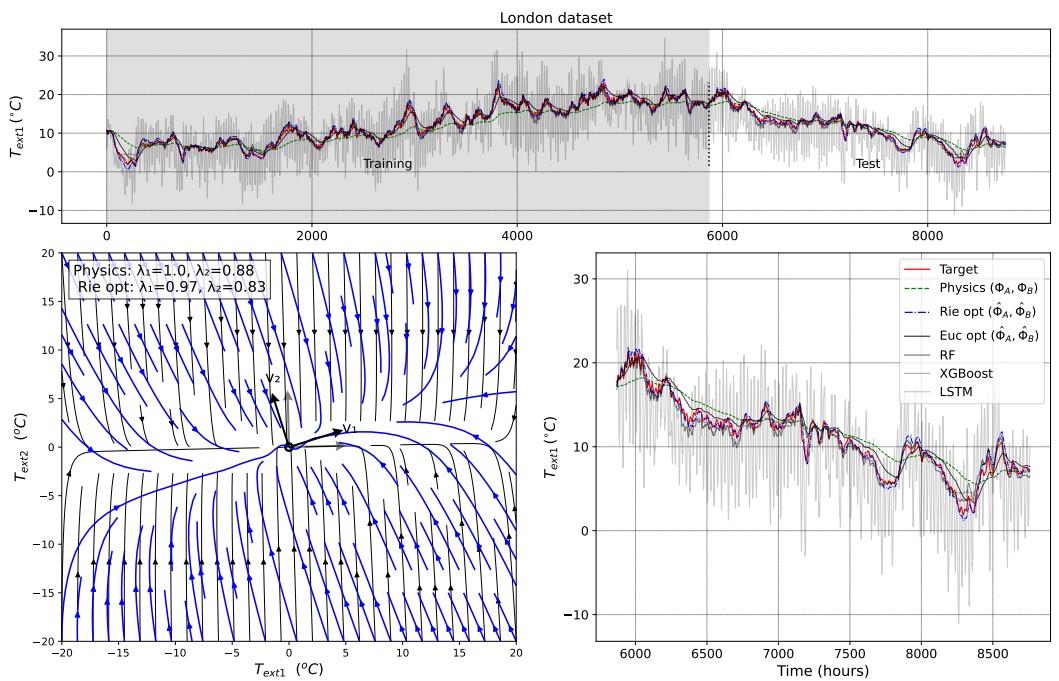

Figure 5: Model training and testing on the London dataset.

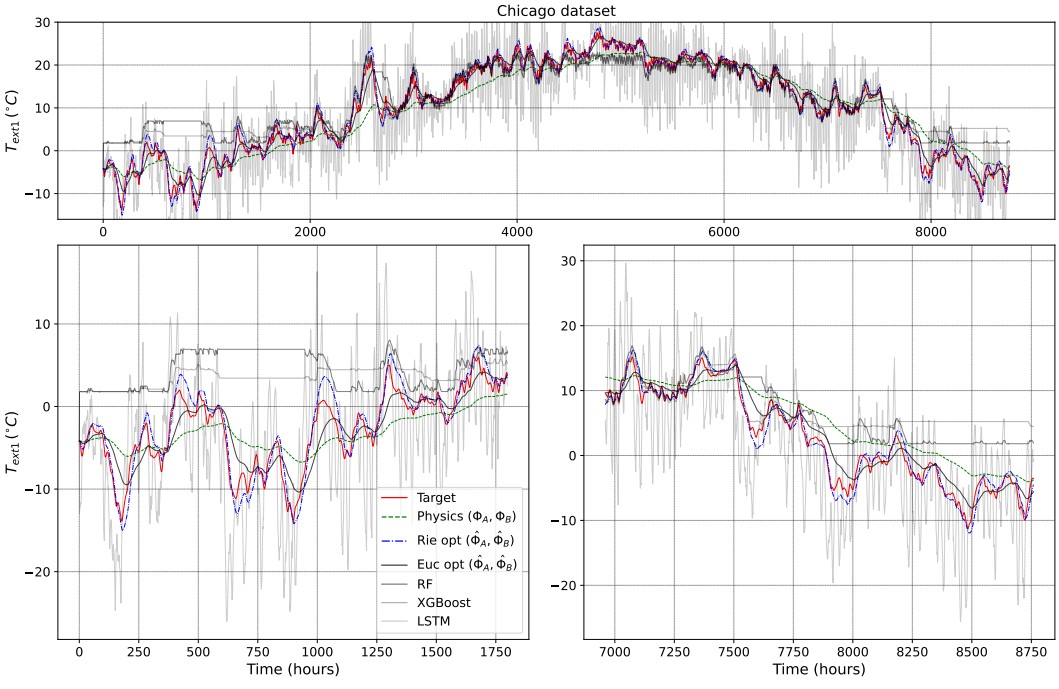

Figure 6: Model testing on the unseen Chicago dataset.

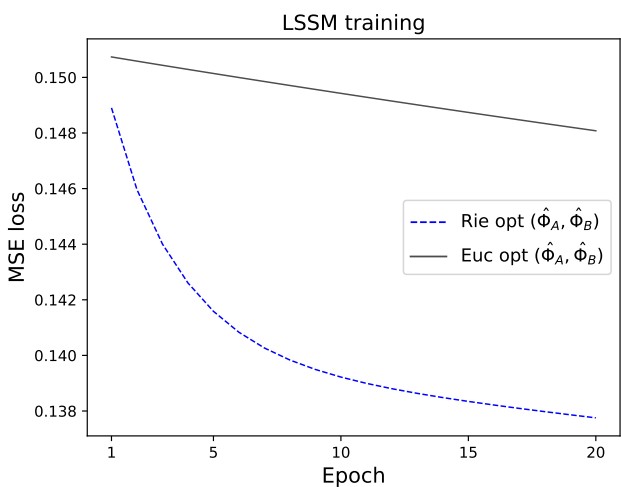

Figure 7: Convergence MSE loss per epoch for optimized LSSMs.

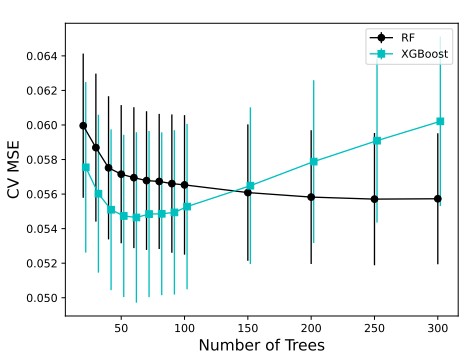

(a) Five-fold cross validated MSE, for a sweep of forest sizes.

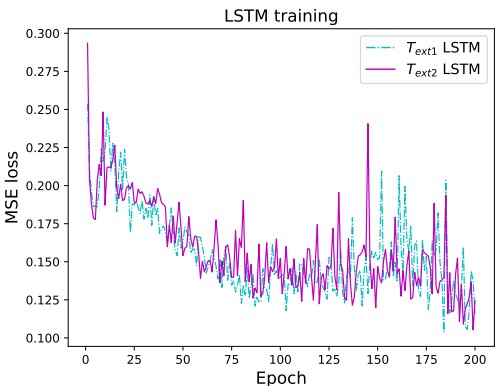

(b) MSE loss per epoch for the LSTMs trained on $T_{ext1}$ and $T_{ext2}$.

Figure 8: Comparison of training convergence across struture-naive models.

# C APPENDIX: CONSERVATIVE USE-CASE SUPPLEMENTARY DETAILS

The left panel of Figure 9 displays plots for the convergence of the training loss with epoch for the SHNN, NeuralODE, and LSTM when applied to the FPU-$\alpha$ system training data. The centre panel indicates how the performance of the models on a $1,000$ steps autoregressive roll-out prediction of test set varies during training. Lastly, the right panel of Figure 9 displays the RMD energy drift for the methods during training, with the energy drift of the SHNN significantly lower than the NeuralODE and LSTM.

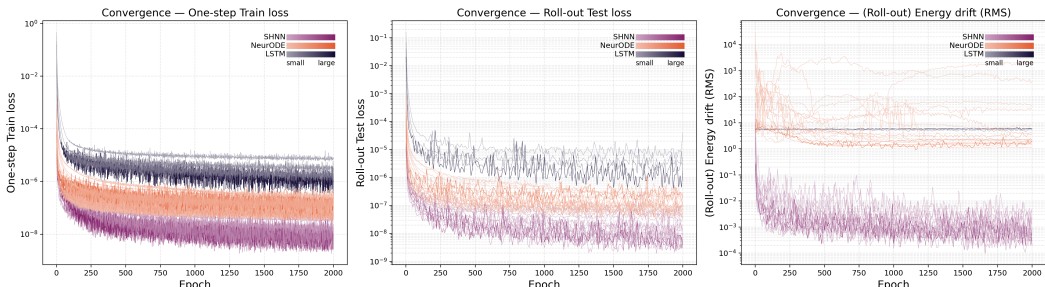

Figure 9: Convergence losses per epoch for SHNN, LSTM and NeurODE across the varying model sizes.

