# OpenReview forum: "Structure-Preserving Machine Learning of Dynamical Systems: A Case for Smaller Models"
_ICLR.cc/2026/Conference — Submitted to ICLR 2026_

### Official Review · Reviewer_dZYj · 2025-10-22

**Soundness:** 2
**Presentation:** 2
**Contribution:** 1
**Rating:** 2
**Confidence:** 5

**Summary:**

This paper studies the influence of structure preservation inductive biases over black box machine learning techniques in dynamical system learning. For the dissipative case, they use Riemannian optimization to ensure the SDP structure of the learnt integrator. For the conservative case, they use symplectic Hamiltonian neural networks to learn the Hamiltonian of the system joint to a symplectic integrator, ensuring stable rollouts. The structure-preserving techniques are compared across different machine learning techniques, such as NeuralODE, LSTM and classic ML (random forest, XGBoost), showing better generalization and the importance of structure identification.

**Strengths:**

* The use of Riemannian optimization is interesting. It is the fist time I have encountered an alternative to parametrizing the Cholesky factor of the friction matrix.
* The figures are clear and represent well the studied systems.

**Weaknesses:**

* The paper lacks a clearly defined aim. The systems studied are very small, and the work mainly combines previously published techniques without offering a substantial methodological or conceptual contribution beyond the analysis of the results.
* The paper is not easy to follow. While the mathematical explanations are clear, the overall structure lacks a consistent narrative. Section 2 introduces the equations and mathematical background for the conservative and dissipative cases, but the text then jumps abruptly to the results, presented in a seemingly disordered manner.
* The paper does not include a clear statement of contributions or significance that would justify its suitability for the present venue.

**Questions:**

* Eq 2: What are the variables U_ext1,ext2?
* Line 190: What is the influence of the misspecification of the physical parameters in the results?
* Line 260: Why 18 dimensions? This comes from M=9 interior degrees of freedom, thus resulting in 2M=18 dimenions of the canonical phase space?
* Section 3.1: I don't fully understand the problem setting. Each dataset correspond to the temperature time series of London and Chicago, each one of dimension R^N_snapsx1. The ambient dry bulb temperature acts as a forcing, which lies on R^N_snapsx2. This is inconsistent with Eq. 2 where the state vector is formed by 2 dimensions (T_ext1, T_ext2) and the forcing vector as 1 dimension (T_ext). Are the two temperatures of London and Chicago concatenated to assemble the full state vector? The problem setting needs to be rewritten, its current form is very misleading.
* Figure 5: What's the meaning of "Physics (\phi_A, \phi_B)"?
* Figure 5: The gray color line of RF, XGBoost and LSTM are almost the same. I´d suggest to use different line styles rather than different colors.
* Line 369: Why is the LSTM only tested for 1 hidden layer? That's too shallow to even be considered deep learning.
* Table 2: I'd suggest to define an objective metric to select the best performance model. "Hand-picked ’best’ size vs. loss" is a very vague description. Why would you discard bigger models which clearly outperform the chosen one? Technically the best option is (L=4, W=144) with only 65k parameters. Why is it discarded?

Minor typos:
* Line 127: It references directly to Eq. 16, but I think it might refer to Eq. 2.
* Line 154: "and which" might refer to "which".
* Line 173-175: Some incorrect parenthesis are added after <0, >0 and >=0.
* Eq. 10: A summation sign (or a parenthesis) is missing in the last term of the Hamiltonian.
* Table 3 caption: "optimisation" might refer to "optimization".

**Details Of Ethics Concerns:**

I have no ethics concerns.

---

### Official Review · Reviewer_UpSz · 2025-10-24

**Soundness:** 2
**Presentation:** 2
**Contribution:** 1
**Rating:** 2
**Confidence:** 3

**Summary:**

The paper studies how incorporating appropriate geometric inductive biases into models for physical dynamical systems can improve parameter efficiency, generalization, and long-term predictive stability, when compared to structure-agnostic methods.

To support the claim empirically, two cases are studied:

(1) a linear 2D dissipative heat-transfer system. To infer the symmetric positive-definite (SPD) system matrix of the corresponding state-space model, Riemannian optimization on the SPD manifold is compared to standard Euclidean regression and other baselines (decision trees, LSTMs).

(2) a conservative Fermi-Pasta-Ulam-Tsingou (FPUT) chain with 18-dimensional phase space. Here, a Symplectic Hamiltonian Neural Network (SHNN) is compared to neural baselines (neural ODE, LSTM) that do not preserve the system’s energy-preserving symplectic structure.

Across both examples, the results show that explicitly encoding geometric structure yields smaller models with improved generalization and prediction accuracy.

**Strengths:**

The paper’s motivation and discussion of the geometric structure of the two example systems is generally consistent with established literature on geometry-aware learning of dynamical systems. Based hereon, suitable structure-preserving methods for each system were selected. As the reported empirical results are in line with prior works in the field, the work serves some illustrative purpose.

**Weaknesses:**

The paper provides limited novelty. While it does not claim methodological or theoretical contributions, the comparative claim (”geometric inductive biases improve parameter efficiency and generalization”) is well-established in literature, and the empirical evaluation confirms known results:

- The heat-transfer system shows a relatively uncomplicated dynamical system (2-dimensional, linear dynamics, ground truth knowledge of the dissipative forces). The formulation of the optimization objective in equations (5)-(7) boils down to a regression problem with SPD constraint on the optimization variable. The benefits of Riemannian optimization methods in similar scenarios are well-understood [1-4], including for optimizing parameters of dynamical systems [5].

- The FPUT chain is a more challenging system (nonlinear elasticity and higher dimensionality), yet similar in nature to prior studies that applied symplecticity-preserving neural networks to mechanical systems, demonstrating increased expressivity over structure-less approaches [6-8].

Common difficulties when learning physical systems [9] (e.g. high-dimensional, partially observed, noisy real-world data) are not considered.
The two different geometry-preserving approaches are selected manually without a generalizable strategy for model selection.
Standard performance metrics (test MSE on seen and unseen data, energy drift) are reported, but no further analytical insights on stability or generalization are provided.
Experimental details are incomplete, including statistical significance (e.g. number of random seeds), hyperparameters for the baseline architectures, and details on generation of testing trajectories for the FPUT system (number of recordings, distribution of initial conditions, value of parameter $\alpha$).

Overall, the experiments offer limited empirical value, by reproducing, instead of advancing, the current understanding of the problem.



**references:**

[1] Gao, Z., Wu, Y., Jia, Y., & Harandi, M. Learning to optimize on SPD manifolds. *Proceedings of the IEEE/CVF Conference on Computer Vision and Pattern Recognition*, 2020.

[2] Huang, Z., & Van Gool, L. A Riemannian network for SPD matrix learning. *Proceedings of the AAAI Conference on Artificial Intelligence*, 2017.

[3] Feragen, A., & Fuster, A. Geometries and interpolations for symmetric positive definite matrices. *Modeling, Analysis, and Visualization of Anisotropy* (pp. 85–113). Springer International Publishing, 2017.

[4] Pennec, X., Fillard, P., & Ayache, N. A Riemannian framework for tensor computing. *International Journal of Computer Vision, 66*(1), 41–66, 2006.

[5] Friedl, K., Jaquier, N., Lundell, J., Asfour, T. & Kragic, D. A Riemannian framework for learning reduced-order Lagrangian dynamics. *International Conference on Learning Representations* (ICLR), 2025.

[6] Gao, Y., Geng, R., Kevrekidis, P., Zhang, H. K., & Zu, J. α-separable graph Hamiltonian network: A robust model for learning particle interactions in lattice systems. *Physical Review E, 111*(1), 015309, 2025.

[7] Finzi, M., Wang, K. A., & Wilson, A. G. Simplifying Hamiltonian and Lagrangian neural networks via explicit constraints. *Advances in Neural Information Processing Systems* (NeurIPS), 2020.

[8] Chen, Z., Zhang, J., Arjovsky, M., & Bottou, L. Symplectic recurrent neural networks. *International Conference on Learning Representations* (ICLR), 2020.

[9] Wang, R., & Yu, R. Physics-guided deep learning for dynamical systems: A survey. *ACM Computing Surveys*, 2021.

**Questions:**

Q1. How does the work position itself, in terms of novelty or analytic depth, relative to existing experiments on geometry-preserving models?

Q2. Since the two experiments employ different structure-preserving models: Is there an overarching principle for selecting appropriate geometric constraints for new systems? E.g., is the presented SHNN suitable for most types of conservative physical systems? Could the intended conceptual difference between *identifying* and *modeling* system dynamics be precised?

Q3. Are there any specific future research questions or directions emerging from this study?

Q4. In Table 1, how was the “LSSM from Physics” case obtained? The description appears ambiguous, as if it was equal to the ground-truth model. Was this the initial guess for the Riemannian and Euclidean optimization runs?

Q5. How does the gained parameter efficiency relate to model complexity, i.e. what is the computational cost of the structure-preserving methods relative to the baselines?

Q6. Could additional details on baseline architectures, hyperparameter selection, and statistical significance be added?

Q7. For the FPUT experiments: Could the number and distribution of test trajectories, the selection of initial conditions, and the selected value of the system parameter α be clarified?

Q8. Could there be additional evidence to support the claims of improved generalization or long-term stability? Such as analysis beyond the reported standard metrics, e.g. comparison of the converged eigenvalues, or reproduction on a more complex system?

---

### Official Review · Reviewer_VGQH · 2025-10-28

**Soundness:** 3
**Presentation:** 2
**Contribution:** 1
**Rating:** 2
**Confidence:** 5

**Summary:**

The paper addresses the well-known problem of showing that small, physically informed models can outperform large, uninformed models in predicting the dynamics of physical systems. Indeed, the adherence of the investigated dynamical systems to mathematical laws constrains the evolution of state variables to geometric manifolds. Embedding this geometric structure into neural model architectures ensures, by design, that the evolution remains confined to these manifolds, even with limited data.

The paper supports this claim in the experimental section, where a satisfactory number of uninformed models are compared with two informed, state-of-the-art approaches: the Riemannian adaptive optimization method (RAdam) [Becigneul & Ganea, 2019] and the symplectic Hamiltonian neural networks (SHNN) [David & Mehats, 2023].

**Strengths:**

As detailed in the Weaknesses section, I would not include originality and significance among the strengths of the paper.

As strong points, I would instead highlight the clarity in presenting the main problem and the detailed reporting of the implementation and training performance of all methods analyzed in the experimental section. The related works are presented briefly but effectively. The two applications are explained in great detail, perhaps excessively, given that they are drawn from the literature.

The experiments have been performed rigorously, and the geometric concepts and interpretations presented in the paper are well-founded.

**Weaknesses:**

I am afraid that the novel contribution of this paper is limited and marginal. Only existing structure-preserving methods are considered (at least they are compared with baselines that differ from those in the original studies). Moreover, the conservative case study lacks originality, as it focuses on learning a synthetic Hamiltonian system, a standard proof-of-concept widely used in the field. In addition, the images chosen to illustrate the results are unclear: they show an arbitrary 2D section of the 18-degree-of-freedom system dynamics, but this perspective does not clarify the points made in the text (I provide more details in the questions to the authors).

The only tangible contribution I can identify is, in the non-conservative case study, the geometric interpretation of the unforced dynamics matrix $A$ as an element of the Symmetric-Positive-Definite manifold, used to justify the application of a structure-preserving learning method. However, this contribution, which is the only addition by the authors since the use case is entirely borrowed from [Xuereb Conti et al. (2023)], is highly specific to the chosen scenario. While there is some discussion of the implications of this interpretation for the stability of the dynamics (i.e., the mapping between the s-plane and z-plane), this treatment lacks further theoretical development for a more general or formal framework and is therefore marginal.

In conclusion, I believe that the paper, in its current form, does not offer a sufficient novel contribution. In particular,

(1) the work only demonstrates that embedding structure in models can help in certain circumstances, a point already established by existing literature on structure-preserving methods [Becigneul & Ganea, 2019][David & Mehats, 2023];

(2) there is no theoretical development that presents these structure-preserving methods from a new perspective;

(3) the use cases do not offer any new insights or interesting applications of the existing models. There is a little geometrical study on the non-conservative use case, but alone it is not sufficient. Papers accepted for their theoretical contributions are typically characterized by more formal treatments, e.g., [Identifying metric structures of deep latent variable models, Stas Syrota et al., 2025][Elucidating Flow Matching ODE Dynamics via Data Geometry and Denoisers, Zhengchao Wan et al., 2025], or by a clear justification of the relevance of newly proposed benchmarks, e.g., [Is Complex Query Answering Really Complex?, Cosimo Gregucci et al., 2025];

(4) a significant portion of the paper is devoted to describing the use cases or applications, but since these are borrowed from other papers, they do not contribute any novelty to the current work.

**Questions:**

### Questions

1. > Considering the discrete nature of physical systems, we must approximate the continuous temperature field in (1) by reducing it to a discrete domain.

   I understand why this system should be discretized for numerical resolution, but I don’t understand why you claim that physical systems have a discrete nature.

2. Figure 2: The top row is clear, but regarding the bottom row, why does the system Hamiltonian still oscillate across different time steps if it is not explicitly a function of time? Is this due to the changes in the other conjugate pairs? Also, is the point you intended to show that the current system state (represented by the cyan dot?) is always tangent to the Hamiltonian white level curves?

3. Figure 2:
   > In the plots, the white contour is the level set at which the invariant energy hypersurface intersects with the $(q_1, p_1)$–slice at time $t$.

   The entire discussion is focused on $(q_4, p_4)$. Is this a typo?

4. Figure 2:
   > When a trajectory is visibly jumping between level sets in these plots, it is indicative of energy drift (non–conservation) arising from model discrepancy from the true energy surface.

   In Figures 2–4, all trajectories seem to jump.

### Suggestions

In Figures 2-4, plotting the Hamiltonian values in the background do not help to highlight the point, but it generates confusion.

The statement at the beginning of the paper,
> parameter perturbations deform the vector field and are observed as changes in flow trajectories on an underlying manifold

and the observation in the non-conservative use case, which interprets the tangent spaces as
> the sets of all possible infinitesimal perturbations of $\Phi_A$ that preserve the symmetric structure

were promising. They appeared to suggest a connection between symplectic and Riemannian geometry, in which symplectic constraints influence geodesics on the manifold, but this idea is not further developed.

---

### Official Review · Reviewer_Kd5k · 2025-10-31

**Soundness:** 3
**Presentation:** 3
**Contribution:** 2
**Rating:** 2
**Confidence:** 3

**Summary:**

The primary goal of this paper is to empirically demonstrate and compare the benefits of incorporating structure-preserving inductive biases for learning the dynamics of both dissipative and conservative physical systems. The key contribution is not a novel algorithm, but a systematic application and validation of existing methods:

For a dissipative system (heat transfer), it applies Riemannian optimization on the Symmetric Positive Definite (SPD) manifold to learn a stable Linear State-Space Model (LSSM).

For a conservative system (FPUT Hamiltonian system), it employs an established Symplectic Hamiltonian Neural Network (SHNN) architecture

The main contribution lies in the comparative experimental framework that shows these structure-aware models—even when smaller—achieve superior generalization, long-term stability, and lower energy drift compared to structure-naive baselines.

**Strengths:**

1. The focus on building efficient, robust, and generalizable models for physical systems is highly relevant for scientific machine learning.

2. The study is strengthened by its clear dichotomy (dissipative vs. conservative) and the inclusion of multiple, popular baseline models.

3. The experimental results robustly support the central claim: structure preservation leads to better long-horizon prediction and stability, especially for unseen initial conditions. The analysis of energy drift in the conservative case is particularly compelling.

4. The paper provides sufficient detail on experimental setup, model sizes, and hyperparameters.

**Weaknesses:**

Although the paper presents strong empirical motivation and exhaustive comparisons for structure-preserving machine-learning modeling, it lacks any substantive algorithmic contribution.
The contribution is limited to "application transfer": no new algorithmic structures, theoretical extensions, or training strategies are proposed. The paper merely benchmarks existing techniques on heat-transfer and FPUT systems and asserts empirical superiority. Such "scenario validation" work is better suited to application-oriented journals, and the reviewer is not sure it meets ICLR's expectations for algorithmic originality and theoretical depth.

**Questions:**

See weakness above.

---

### Meta-Review · Area_Chair_wahm · 2025-12-16

**Summary:**

The reviewers agree that that the work lacks meaningful novelty. The methods used are combinations of existing structure-preserving approaches, and the contributions are limited to applying them to two small systems. Reviewers also note that much of the paper focuses on describing known use cases, and that the experimental findings mainly confirm results already established in prior literature.

**Reviewer Concerns:**

Since no rebuttal was provided, none of the reviewers concerns were addressed. All major points raised by the reviewers remain outstanding.

**Reviewer Scores:**

Because no rebuttal was submitted and the reviewers were consistent in their critiques, it is unlikely that any of them would change their score.

---

### Decision · Program_Chairs · 2026-01-26

Reject